

# High-resolution modeling of thermal thresholds and environmental influences on coral bleaching for local and regional reef management

Naoki H. Kumagai, Hiroya Yamano and Committee Sango-Map-Project

Center for Environmental Biology and Ecosystem Studies, National Institute for Environmental Studies, Tsukuba, Ibaraki, Japan

Corresponding author
Naoki H. Kumagai,
nh.kuma@gmail.com

## ABSTRACT

Coral reefs are one of the world's most threatened ecosystems, with global and local stressors contributing to their decline. Excessive sea-surface temperatures (SSTs) can cause coral bleaching, resulting in coral death and decreases in coral cover. A SST threshold of 1 °C over the climatological maximum is widely used to predict coral bleaching. In this study, we refined thermal indices predicting coral bleaching at high-spatial resolution (1 km) by statistically optimizing thermal thresholds, as well as considering other environmental influences on bleaching such as ultraviolet (UV) radiation, water turbidity, and cooling effects. We used a coral bleaching dataset derived from the web-based monitoring system Sango Map Project, at scales appropriate for the local and regional conservation of Japanese coral reefs. We recorded coral bleaching events in the years 2004–2016 in Japan. We revealed the influence of multiple factors on the ability to predict coral bleaching, including selection of thermal indices, statistical optimization of thermal thresholds, quantification of multiple environmental influences, and use of multiple modeling methods (generalized linear models and random forests). After optimization, differences in predictive ability among thermal indices were negligible. Thermal index, UV radiation, water turbidity, and cooling effects were important predictors of the occurrence of coral bleaching. Predictions based on the best model revealed that coral reefs in Japan have experienced recent and widespread bleaching. A practical method to reduce bleaching frequency by screening UV radiation was also demonstrated in this paper.

## INTRODUCTION

Biological communities can shift toward alternative stable states in response to changing climate (*Parmesan & Yohe, 2003*). Coral reefs are one of the most susceptible ecosystems to global warming and local environmental stressors (*Hoegh-Guldberg, 1999*; *West & Salm, 2003*). Rising sea-surface temperatures (SST) can cause bleaching in reef-building corals, especially during summer (*Hoegh-Guldberg, 1999*; *Brown et al., 2002*; *West & Salm, 2003*).

Excessive thermal stress leads to expulsion, digestion, or reduced pigmentation of symbiotic dinoflagellate algae in coral cells, resulting in the exposure of white coral skeletons (i.e., bleaching; *Hoegh-Guldberg, 1999*; *Brown et al., 2002*). Prolonged warming trends in sea temperature have been predicted to increase the frequency and severity of bleaching in the future, leading to mass mortality of corals (*Hoegh-Guldberg, 1999*; *Donner et al., 2005*; *Donner, 2009*; *McClanahan, Maina & Ateweberhan, 2015*). Reef management relies on not only global measures to reduce climate warming but also local measures to control environmental influences on coral resilience (*West & Salm, 2003*). Spatial and temporal predictions of coral bleaching under varying environmental conditions could therefore provide valuable information to support local management of coral reefs.

The degree heating week (DHW) index of cumulative thermal stress, developed by the National Oceanic and Atmospheric Administration Coral Reef Watch (NOAA CRW), has been widely used to predict coral bleaching. DHW is based on SST derived from satellite images, and is computed as the sum over a period of 12 weeks of thermal stress exceeding 1 °C above historical summer monthly SST (*Liu, Strong & Skirving, 2003*). DHW over 4 °C weeks indicate severe coral bleaching and constitute a bleaching alert threshold (*Liu, Strong & Skirving, 2003*).

Despite its increasing use globally, the predictive performance of DHW may not be sufficient for local reef management, as DHW on average detects only 40% of global coral bleaching events (*Donner, 2011*). This low predictive performance may be due to the use of a fixed thermal threshold of 1 °C above baseline SST. Previous studies have suggested that thermal stress of 1 °C or below can induce coral bleaching (*Brown, 1997*; *McWilliams et al., 2005*; *Kleypas, Danabasoglu & Lough, 2008*). In addition, historical temperature variability can affect bleaching and coral resilience (*Brown et al., 2002*; *West & Salm, 2003*). As a consequence, some studies have used modified indices, such as the sum of thermal stress over 0 °C above baseline SST (*Yee, Santavy & Barron, 2008*; *Kayanne, 2017*). *Donner (2011)* proposed two-modified DHW indices: an index using historical SST variability as the bleaching alert threshold, and an index using the mean of the warmest monthly SST of each year as the baseline SST.

To evaluate the effects of global and local stressors on corals, a high-performance predictive model operating at high-spatial resolution is required. Global stressors such as thermal stress can vary at a local scale (*Strong et al., 2002*; *Liu et al., 2014*). Furthermore, there are potentially interacting environmental stressors such as ultraviolet (UV) radiation (*Hoegh-Guldberg, 1999*; *West & Salm, 2003*; *Maina et al., 2008*; *Yee, Santavy & Barron, 2008*) and variables such as water turbidity (*West & Salm, 2003*; *Oxenford & Vallés, 2016*), topography of the sea floor (*West & Salm, 2003*; *Oliver, Berkelmans & Eakin, 2009*), and exposure to winds (*West & Salm, 2003*) and currents (*Nakamura & van Woesik, 2001*; *West & Salm, 2003*) that can affect coral bleaching. For example, increasing the speed of surface currents and winds can reduce bleaching risk by increasing mixing in surface seawater (*Nakamura & van Woesik, 2001*; *Maina et al., 2008*).

Modeling coral bleaching at a local scale also requires high-resolution observational records, as omission of bleaching events can lead to poor predictive power in models (*Oliver, Berkelmans & Eakin, 2009*). ReefBase (*Tupper et al., 2011*) and the Bleaching

Database V1.0 (*Donner, Rickbeil & Heron, 2017*) provide high record coverage in some areas, including in the Great Barrier Reef and the Caribbean (*van Hooidonk & Huber, 2009*). However, records are still limited for other areas, such as the Pacific islands (*Donner, Rickbeil & Heron, 2017*). One possible reason for this data gap is language barrier. A considerable amount of data in ReefBase (*Tupper et al., 2011*) have been provided by nonprofessional (citizen) specialists who are not native English speakers. Few Japanese records ($N \leq 64$) are found in the global databases, despite the large amounts of research conducted on coral reefs in Japan. To collect and collate observational records of corals throughout Japan, diverse Japanese stakeholders, including professional scientists, government officials, and citizens, constructed a web-based monitoring system for Japanese coral reefs in 2008, the Sango Map Project (*Namizaki et al., 2013*). Collecting observational records in a web-based database proved to be effective in Japan, as internet service is available to the vast majority of the population. In addition, the use of Japanese language allowed a larger number of stakeholders to contribute, including stakeholders from populated islands where diving services are available. This project contributed key data to the International Year of the Reef Year in Review report (*Staub & Chhay, 2009*).

In this study, we aimed to improve predictive power in models of coral bleaching at high-spatial resolution, in order to inform local and regional reef management. We used observational records of coral bleaching derived from the Sango Map Project, and we compared the predictive performance of multiple thermal indices and their modifications in models with multiple explanatory variables. We developed a novel derivation of DHW (hereafter "filtering threshold") to compute thermal stress below 1 °C in excess of the baseline SST, using historical SST variability as a threshold. We optimized the filtering threshold by statistical estimation of each type of DHW and degree heating month (DHM) index. To maximize predictive performance, we then optimized the combination of multiple explanatory variables while optimizing the filtering threshold. Based on the model with maximum predictive performance, we produced spatial predictions of coral bleaching in the study area, as well as predictions under reduced local environmental stresses. Our results provide a reference for local reef management in Japan, although our methods could be applied for local reef management in other areas.

## MATERIALS AND METHODS

### Observational records of coral bleaching

We used observations from the Japanese coasts submitted to the Sango Map Project (http://www.sangomap.jp/) up to March 2017. Observations were composed of the following information: (1) presence or absence of corals; (2) longitude and latitude of the location, searchable through the Google Maps API (https://developers.google.com/maps/); (3) name of the location; (4) date, month, and year of the observation; (5) method of survey (scuba diving, snorkeling, glass boat, walking, or other); (6) water depth in meters; (7) observer's professional background (professional scientist, nonprofit or nongovernmental organization, tourism, or other); (8) level of severity of coral bleaching (high, medium, low, nonbleaching, or not available) derived from the bleaching dataset in

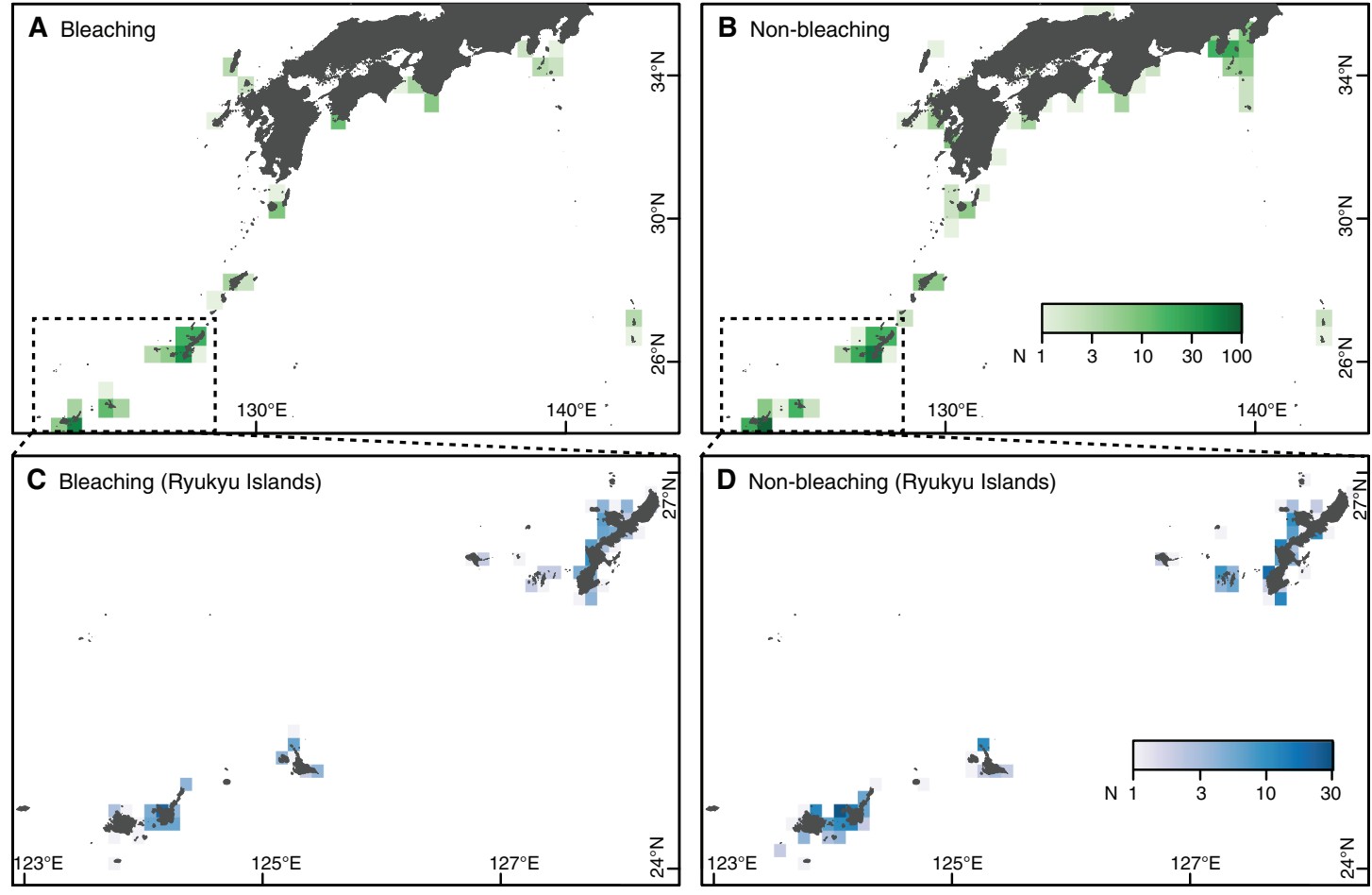

**Figure 1 Study area and number of observations in southern Japan.** (A, B) Whole study area, with the main study area enclosed by a dashed square. (C, D) Main study area: Ryukyu Islands. (A, C) Observations of coral bleaching. (B, D) Observations of nonbleaching. Japanese map is publicly available from the *Geospatial Information Authority of Japan (2015)* (http://www.gsi.go.jp/ENGLISH/index.html).

ReefBase. We confirmed or rejected questionable records, such as observations made on land or in the open ocean and observations of doubtful coral species.

After quality control and exclusion of records lacking information on bleaching, we obtained 668 independent records between July 2004 and October 2016. Of these observations, 52 were submitted by professional scientists, 152 by nonprofit or nongovernmental organizations, and 134 by tourists. Fifty-nine observations were conducted as part of CoralWatch (http://www.coralwatch.org/) and 63 as part of ReefCheck Japan (http://www.reefcheck.jp/). The records provided good spatial coverage of coral reefs in Japan (Fig. 1). Most of the records were obtained in the first three years following the launch of the Sango Map Project, including 449 records from 2008 to 2010 alone. In addition, 82 and 111 records were reported in 2013 and 2016, respectively, when mass bleaching events were observed throughout Japan (*Kayanne, 2017*; *Kayanne, Suzuki & Liu, 2017*).

Records of bleaching not induced by thermal stress were regarded as nonbleaching observations for the purpose of this study. We therefore reclassified 107 bleaching observations as nonbleaching observations (Step 1 in Table 1). Following screening, the prevalence of records was more biased than prior to screening, with 228 bleaching and 440 nonbleaching observations collated (Fig. 1). However, the risk of biased predictions was still deemed low (Step 2 in Table 1). Annual and spatial patterns of bleaching occurrences were consistent with those reported previously in Japan (*Kayanne, 2017*; *Kayanne, Suzuki & Liu, 2017*). We assessed spatial autocorrelation in the residuals of the prediction model of the NOAA CRW DHW (Step 2 in Table 1), using the spatial autocorrelation coefficient (Moran's *I*). We confirmed that there was no significant autocorrelation in the residuals, indicating no significant spatial bias in the data (*Dormann et al., 2007*).

## Thermal indices

To calculate thermal indices, we used daily data at a spatial resolution of 1 km (0.01°) from the Multi-scale Ultra-high Resolution Sea Surface Temperature (MUR SST) Analysis version 4.1 (*JPL MUR MEaSUREs Project, 2015*) (http://dx.doi.org/10.5067/GHGMR-4FJ04). The MUR SST product is a blend of SST from six satellites and thus provides higher accuracy than single-satellite products. MUR SST data are only available from 2002 to May 2017, an insufficient period of time to calculate maximum monthly mean (MMM) climatology (Table 2). In addition to data from the MUR SST, we also used data for the years 1985–2002 from the Optimum Interpolation SST (OI SST) version 2 (*Reynolds et al., 2007*) (http://www.esrl.noaa.gov/psd/data/gridded/data.noaa.oisst.v2.highres.html). To correct the SST bias between MUR SST and OI SST, we added the bias in the 2002–2017 monthly climatology to the OI SST data, after down-scaling to 0.01° using inverse distance weighting interpolation (*Tabor & Williams, 2010*; *Yara et al., 2011*).

Using the monthly mean SST from 1985 to 2015, we obtained two types of MMM climatologies. The first MMM climatology follows the NOAA CRW version 3 protocol (*Liu et al., 2014*, *2017*). The temporal midpoint was recentered to that of the heritage 50 km MMM (1985–1990, 1993) using the approach of *Heron et al. (2014)* as follows:

$$\mathrm{SST}_{\mathrm{recentered}i} = \mathrm{SST}_i - \mathrm{slope}_i \times (T_{1985-2015} - T_{1985-1993}),$$

where $\mathrm{SST}_i$ is the SST climatology as obtained above and $\mathrm{SST}_{\mathrm{recentered}i}$ is the recentered SST climatology at cell *i*. The linear trend of monthly mean SST between the center times of the two-time durations ($T_{1985-2015}$, $T_{1985-1993}$) is represented by $\mathrm{slope}_i$ at cell *i*. The down-scaled and recentered MMM correlated significantly with the CRW MMM version 3 (Figs. S1 and S2).

The second climatology, known as $\mathrm{MMM}_{\mathrm{max}}$ climatology (*Donner, 2009*, *2011*), uses the mean of the warmest month of each year instead of the mean of the warmest month in the climatological years in the MMM climatology (Table 2). The warmest month is not always the same among years and, therefore, $\mathrm{MMM}_{\mathrm{max}}$ is larger than MMM (Table 2; Figs. S3A–S3C) and represents the seasonal peak in SST more accurately than MMM

**Table 1 Flowchart summarizing the three steps in our analysis.**

| | Procedure | Approach | Reference |
|---|---|---|---|
| **Step 1** | Control of observation errors | | |
| | Excluding bleaching events not induced by thermal stress | Observation records of small bleaching events (e.g., those within microatolls, or caused by disease or predation) and observations made after the small bleaching event were regarded as nonbleaching if the 1 km resolution DHW value at observation site did not exceed zero | This study (2018) |
| **Step 2** | Assumptions for observed data | | |
| | Checking equality in observations of occurrence and absence of bleaching, where higher prevalence (usually biased to occurrences) results in larger predicted probabilities (i.e., biased predictions) | Using an evaluation index that is less dependent on prevalence (TSS). The evaluation threshold was also optimized (see Step 4) | *Allouche, Tsoar & Kadmon (2006)* and *Liu et al. (2005)* |
| | Avoiding spatial autocorrelation in the data, which can increase false-positive predictions | Evaluating the spatial autocorrelation coefficient (Moran's $I$) of residuals from a prediction model. If residuals are spatially biased, spatially clustered data should be filtered | *Dormann et al. (2007)* and *Boria et al. (2014)* |
| **Step 3** | Assumptions for environmental variables | | |
| | Screening correlated environmental variables | If correlations between variables are high ($|r| > 0.7$), correlated variables should be excluded to reduce multicollinearity, which can affect both GLM and RF | *Dormann et al. (2013)* |
| **Step 4** | Evaluation and model assessment | Multiple performance metrics were used to avoid Type I and Type II errors. Models using standard and optimized thresholds were assessed. A statistical model (GLM) and a machine learning model (RF) were used | *Zuur et al. (2009)* |
| | Optimizing combinations of explanatory variables | Statistical selection of a subset of explanatory variables from all variables (thermal index and six other variables) to maximize TSS. The two most influential variables (DCW and UV-B) were always included and, therefore, 15 possible combinations of the other variables were evaluated | *Zuur et al. (2009)* |
| | Optimizing the evaluation threshold | Optimizing the threshold to discriminate occurrence and absence from the predicted probability of bleaching. Although statistical models predicting occurrence or absence typically output results as probabilities, using a 0.5 (i.e., midpoint) threshold can yield biased results under unequal class prevalence. To avoid this problem, TPR–TNR sum maximization was used to optimize the threshold (Table 2) | *Manel, Williams & Ormerod (2001)* and *Liu et al. (2005)* |
| | Optimizing the filtering threshold | To optimize DHW and DHM, the filtering threshold was adjusted by 0.01 °C of precision to maximize predictive power (i.e., TSS) for each combination of explanatory variables | This study (2018) |
| | Evaluation using 10-fold cross-validation | A randomly selected 30% subset of the data were used as testing data, and the remaining data were used as training data. Prediction models were built with the training data and evaluated against the testing data. The test was repeated 10 times for each filtering threshold and combination of explanatory variables | *Hijmans et al. (2017)* |

| | Procedure | Approach | Reference |
|---|---|---|---|
| | **Table 1 (continued).** | | |
| Step 5 | Coral bleaching prediction | | |
| | Prediction under observed environmental conditions | Using the best performing model built in each cross-validation, the probability of coral bleaching was predicted for the study area | *Hijmans et al. (2017)* |
| | Prediction under reduced UV radiation due to screening effect | Coral bleaching frequency may be reduced by a 40% reduction in UV radiation and a 40% increase in water turbidity due to screening | *Cacciapaglia & van Woesik (2016)* |

**Notes:**
Steps 1–3: assessment of the validity of assumptions for explanatory variables and data, respectively. Step 4: evaluation of predictive models. Step 5: predictions of coral bleaching.
DCW, degree cooling week; DHM, degree heating month; DHW, degree heating week; RF, random forest; TNR, true negative rate; TPR, true positive rate; TSS, true skill statistic; UV, ultraviolet.

climatology. This method is particularly effective in tropical zones with reduced seasonality (*Donner, 2011*).

We calculated eight types of thermal and cooling indices for each grid cell and observation day, including mean weekly and monthly SST, DHM, DHW (MMM + $\alpha$ °C), DHW (MMM + $\alpha$ °C) using SST variation as the bleaching alert threshold, DHW (MMM$_{max}$ + $\alpha$ °C), DHW (MMM + $\beta\sigma_m$ °C), and degree cooling weeks (DCW) (see Table 2 for a detailed derivation of the indices). Historical SST variability ($\sigma_m$) (Table 2) was calculated with the monthly mean SST from 1985 to 2015, and ranged from 0.36 to 0.71 with a median of 0.57 (Fig. S2D). Although DCW is calculated with a similar algorithm to that used for DHW, DCW was not significantly correlated with DHW. We therefore included DCW as a covariate in our models. The filtering thresholds ($\alpha$ and $\beta$) were fixed to 1 in the standard thermal indices and optimized in our indices.

## Additional environmental variables

Monthly UV-B and PAR data were obtained from the Japan Aerospace eXploration Agency Satellite Monitoring for Environmental Studies (JASMES; http://kuroshio.eorc.jaxa.jp/JASMES/; accessed 25 June 2017) and derived from the average of data extracted from the Aqua and Terra sensors of moderate resolution imaging spectroradiometer (MODIS; http://modis.gsfc.nasa.gov/data/dataprod/). Although both UV-B and PAR may affect coral bleaching (*Hoegh-Guldberg, 1999*), the variables were significantly correlated ($r = 0.79$) (*Yee, Santavy & Barron, 2008*). We excluded PAR from our analysis (Step 3 in Table 1) as parameters may be misestimated in statistical modelings and machine learnings under multicollinearity (*Dormann et al., 2013*).

To quantify the speed of surface currents, we extracted data from the HYCOM+ NCODA Global 1/12° Analysis GLBu0.08 from 1997 to 2017 (https://hycom.org/dataserver/gofs-3pt0/analysis/; accessed 22 June 2017). We obtained climatological median from July to September, the months during which most of the recorded bleaching events occurred. To quantify wind speed, typhoon tracking data were obtained from the Regional Specialized Meteorological Center Tokyo (http://www.jma.go.jp/jma/jma-eng/jma-center/rsmc-hp-pub-eg/trackarchives.html; accessed 22 June 2017). We calculated the wind speed index for each grid cell as the length of time without typhoons, with

**Table 2 Summary of indices and methods used in this study.**

| Terminology | Definition | Interpretation | Reference |
|---|---|---|---|
| Monthly sea-surface temperature (SST) | Bleaching alert threshold: >30 °C | Simple indices for coral bleaching | *Guinotte, Buddemeier & Kleypas (2003)* and *Yara et al. (2011)* |
| Weekly SST | Bleaching alert threshold: 31.5 °C | Simple indices for coral bleaching | *Kleypas, McManus & Meñez (1999)* |
| Maximum of the monthly mean SST climatology (MMM) | The warmest of the 12 climatological monthly mean temperatures, calculated for each location | Historical baseline temperature (Fig. S3A) | *Liu, Strong & Skirving (2003)*, *Liu et al. (2014, 2017)*, and *Heron et al. (2014)* |
| Mean of the warmest monthly mean SST of each year ($MMM_{max}$) | The mean of the warmest monthly mean of each year during the climatological duration, calculated for each location | Historical baseline temperature, better representing actual warmest temperature than MMM (Fig. S3B) | *Donner (2009, 2011)* |
| HotSpots (HS) | $HS_i = \begin{cases} SST_i - MMM, & \text{if } SST_i > MMM \\ 0, & \text{if } SST_i \leq MMM \end{cases}$ (*i*: time) | Positive only SST anomalies, index of coral bleaching hotspot | *Liu, Strong & Skirving (2003)* and *Liu et al. (2014, 2017)* |
| Historical SST variability ($\sigma_m$) (v) | $\sqrt{\frac{\sum_{i=1}^{n} Max\,mo.SST_i - \overline{Max\,mo.SST}}{n-1}}$  Max mo. $SST_i$: Maximum monthly SST in year *i* in *n* years | Index of interannual variability in maximum monthly SST | *Donner (2011)* |
| Degree Heating Month: DHM (MMM + α °C) | $\sum_{i=1}^{12}(HS_i, \text{if } HS_i \geq \alpha\,°C)$ (*i*: month) Bleaching alert threshold: DHM > α °C; usually α = 1 °C but the threshold was optimized in this study | Index of accumulated thermal stress experienced by corals | *Donner et al. (2005)* |
| NOAA CRW degree heating week: DHW (MMM + α °C); DHW with the bleaching alert of 4 °C | $\frac{1}{7}\sum_{i=1}^{84}(HS_i, \text{if } HS_i \geq \alpha\,°C)$ (*i*: day) Bleaching alert threshold: DHW > 4 °C (Fig. S1B); usually α = 1 °C but the threshold was optimized in this study | Index of accumulated thermal stress experienced by corals | *Liu, Strong & Skirving (2003)* and *Liu et al. (2014, 2017)* |
| Degree heating week: DHW (MMM + 1 °C), DHW using historical SST variability ($\sigma_m$) as the bleaching alert | $\frac{1}{7}\sum_{i=1}^{84}(HS_i, \text{if } HS_i \geq \alpha\,°C)$ (*i*: day) Bleaching alert threshold: DHW > $\sigma_m$/median ($\sigma_m$). The global value of 1/median ($\sigma_m$) = 2.45 °C$^{-1}$, reported by *Donner (2011)* was used | Index of accumulated thermal stress experienced by corals, considering variability of past SST (Fig. S3D) for bleaching alert threshold. For models with multiple explanatory variables, a model including SST variability together with DHW corresponds to this type of DHW | *Donner (2011)* |
| Degree heating week: DHW ($MMM_{max}$ + α °C), DHW using $MMM_{max}$ as the baseline climatology | $\frac{1}{7}\sum_{i=1}^{84}(HS_{max\,i}, \text{if } HS_{max\,i} \geq \alpha\,°C)$ (*i*: day) Bleaching alert threshold: DHW > 4 °C; usually α = 1 but optimized in this study | Index of accumulated thermal stress experienced by corals, exceeding mean of warmest monthly SST in each year | *Donner (2009, 2011)* |
| Degree heating week: DHW (MMM + $\beta\sigma_m$ °C), DHW using the historical SST variability ($\sigma_m$) as the filtering threshold | $\frac{1}{7}\sum_{i=1}^{84}(HS_i, \text{if } HS_i \geq \beta\sigma_m\,°C)$ (*i*: day) Bleaching alert threshold: DHW > 4 °C; conservatively β = 1 but optimized in this study | Index of accumulated thermal stress experienced by corals, considering variability of past SST (Fig. S3D) to assess the filtering threshold | This study (2018) |
| Degree cooling week: DCW (c) | $\frac{1}{7}\sum_{i=1}^{84}(CS_i, \text{if } CS_i \geq 0\,°C)$ (*i*: day) $CS_i$: Cool spots = MMM−$SST_i$; (note that a clear definition is not given in *Jones et al., 2017*) | Index of accumulated reduced thermal stress (cooling effect) experienced by corals | *Jones et al. (2017)* |

| Terminology | Definition | Interpretation | Reference |
|---|---|---|---|
| Water depth (d) | Water depth reported where bleaching or nonbleaching was observed | Depth can affect coral bleaching by reducing thermal stress and light radiation | *Oliver, Berkelmans & Eakin (2009)* |
| Water turbidity (k) | Diffuse attenuation coefficient at 490 nm ($K_{490}$), representing the rate at which light is attenuated with water depth | Turbidity can affect coral bleaching by reducing light stress | *Oliver, Berkelmans & Eakin (2009)* |
| UV-B (u) | Irradiance of ultraviolet radiation ranging from 280 to 315 nm ($Wm^{-2}$) | Strong solar irradiance, particularly from UV, is an important factor affecting coral bleaching through thermal and photochemical damage | *Hoegh-Guldberg (1999)*, *West & Salm (2003)*, and *Maina et al. (2008)* |
| Speed of surface current (s) | sqrt (longitudinal velocity$^2$ + latitudinal velocity$^2$) $ms^{-1}$ | Surface current can reduce bleaching risk by mixing surface water | *Nakamura & van Woesik (2001)* |
| Overall accuracy | (true positives + true negatives)/(total number of predictions) | Proportion of correct predictions allowing a correct prediction with no prediction skill | *Allouche, Tsoar & Kadmon (2006)* |
| True positive rate (TPR) = sensitivity | (true positives)/(true positives + false negatives) | Accuracy of positive predictions (cf. $1 - TPR =$ false negative rate = rate of Type II errors) | *Liu et al. (2005)* |
| True negative rate (TNR) = specificity | (true negatives)/(false positives + true negatives) | Accuracy of negative predictions (cf. $1 - TNR =$ false positive rate = rate of Type I errors) | *Liu et al. (2005)* |
| True skill statistic (TSS) | TPR + TNR − 1 | Index representing prediction power ranging from −1 to 1. A score of 1 indicates perfect prediction, while a score of 0 indicates no prediction skill | *Allouche, Tsoar & Kadmon (2006)* |
| TPR–TNR sum maximization | Maximizing the sum of TPR and TNR (equivalent to maximizing TSS) | Considering both positive and negative predictions equally, prediction skill is expected to be maximized | *Manel, Williams & Ormerod (2001)* and *Liu et al. (2005)* |
| Generalized linear model of binomial response (GLM) | A model fitting data using maximum likelihood that links the response variable (bleaching or nonbleaching) to a linear model via a converting function (logit), assuming a binomial distribution | Parameter coefficients of environmental variables are estimated, to predict the probability of coral bleaching. The optimized model can be described as a formula | *Hijmans et al. (2017)* |
| Random forest (RF) | A machine learning method based on conditional branches of interactions among explanatory variables, created by repeatedly selecting random subsets of the data | The method provides high predictive performance in the form of probabilities. However, predictions cannot be described as an easily communicable formula, but rather are supplied as electronic data | *Breiman (2001)* and *Hijmans et al. (2017)* |

typhoons defined as wind speeds over 15 ms$^{-1}$. However, the wind speed index was strongly correlated with DHW ($r = 0.86$) and therefore was excluded from the analysis.

We used the diffuse attenuation coefficient ($K_{490}$) as an index of water turbidity, which can reduce light radiation stress involved in bleaching (Table 2). A monthly composite of $K_{490}$ (4 km, Level-3 binned MODIS AQUA products) was obtained from the NOAA OceanColor database (https://oceancolor.gsfc.nasa.gov; accessed 8 September 2017) for the months July–September.

Data on current speed and diffuse attenuation were down-scaled to 1 km using bilinear interpolation. When environmental variables were not available for coastal cells, we used inverse distance weighting interpolation to estimate coastal values.

## Model evaluation and optimization

We evaluated coral bleaching models based on the accuracy of both positive (bleaching) and negative (nonbleaching) predictions. Most studies have evaluated models of coral bleaching based only on overall accuracy, such as the proportion of correct predictions and AIC (*Maina et al., 2008*, *2011*; *McClanahan, Maina & Ateweberhan, 2015*; *Kayanne, 2017*; *Welle et al., 2017*), whereas a few studies have differentiated the accuracy of positive and negative predictions (*Yee, Santavy & Barron, 2008*; *van Hooidonk & Huber, 2009*; *Donner, 2011*). When the number of bleaching and nonbleaching observations is unequal (i.e., under class imbalance), model predictions can be biased. We therefore used four evaluation metrics: overall accuracy, true positive rate (TPR), true negative rate (TNR), and true skill statistic (TSS) (defined in Table 2). TSS quantifies prediction skill as the index weighs positive and negative predictions equally (*Allouche, Tsoar & Kadmon, 2006*).

To assess the combined effects of thermal stress and multiple environmental influences on coral bleaching, we constructed prediction models of bleaching with two approaches: generalized linear model (GLM) with a binomial error distribution and a logit link function, and random forests (RFs; *Breiman, 2001*). Although both models compute predictions in the form of probabilities, the models are based on different algorithms. GLM is an extension of regression models, whereas RF is a machine learning method that uses randomly repeating classifications to capture complex interactions among explanatory variables (Table 2). Therefore, GLM has an advantage of which fitted model can be written as a formula that is easy to be used subsequently.

We confirmed that the data met the assumption of binomial GLM (i.e., the residual deviance per degree of freedom was less than 1.5; *Zuur et al., 2009*). GLM was applied with the "glm" function in base R (*R Core Team, 2017*), and RF was applied with the "randomForest" function of the randomForest R package. RF was used under standard settings to avoid overfitting the training data. However, we followed the recommendation of *Hijmans et al. (2017)*, by specifying the model as a "regression model," even though the response variable was categorical. The relative importance of explanatory variables was calculated with the "importance" function of the MuMIn package (*Bartoń, 2015*) for GLM and of the randomForest package for RF (*Liaw & Wiener, 2002*).

Predicted probabilities were transformed into bleaching and nonbleaching categories with the threshold that maximized the sum of TPR and TNR (*Liu et al., 2005*).

The midpoint (i.e., 0.5) is often used as a threshold (Fig. S1C), although it is sensitive to class imbalance in the training data and may, therefore, lead to inaccurate predictions (*Liu et al., 2005*). This issue is addressed in studies of species distribution modeling, although it remains poorly addressed in studies of mass bleaching (*van Hooidonk & Huber, 2009*). We used the "evaluation" function of the dismo R package (*Hijmans et al., 2017*) for evaluation and optimization of the threshold.

Models were evaluated by 10-fold cross-validations using TSS as the evaluation index. In each repeat, we separated 30% of the data as testing data and used the remaining 70% for constructing GLM and RF (Step 4 in Table 1). We optimized the filtering thresholds (Fig. S1A) for DHM and DHWs by cross-validation, while the filtering thresholds were fixed at 1.0 °C in the standard indices (Step 4 in Table 1). We selected the optimum filtering threshold between 0 and 1.5 °C for indices using a constant threshold ($\alpha$), whereas we examined the coefficient of $\sigma_m$ ($\beta$) between 0.1 and 2.5 for indices based on historical variability (Table 1). We conducted optimizations with 0.01 precision for both types of indices, i.e., with 151 and 241 submodels, respectively.

For models with multiple explanatory variables, we considered DCW, historical SST variability, UV-B, water turbidity, water depth, and current speed, in addition to the thermal index. The two most influential variables (DCW and UV-B) were always included in models with multiple explanatory variables (Step 4 in Table 1). The optimum set of explanatory variables was specified through cross-validation. The set of variables that best explain variation in the testing data was selected among all 15 possible combinations. In total, we evaluated 22,650 and 36,150 models (10 cross-validations × 15 variable combinations × 151 or 241 submodels) for each GLM and RF model, respectively.

Finally, we predicted coral bleaching in the warmest month of the main coral habitat in the study area using the best predictive model (Step 5 in Table 1). We also assessed reduction in UV-B as a possible adaptive measure by reducing UV-B radiation by 40% and increasing water turbidity by 40%, thereby simulating the effects of screening with fishnets (Step 5 in Table 1). These levels of changes were consistent with in situ examination in Onna Village in the Ryukyu Islands (*Okinawa Prefecture, 2017*). Predictions were obtained from each model built in the 10 cross-validations, and subsequently averaged among the models. Spatial data were obtained from the Global Map Japan version 2.1 Vector data, provided by the *Geospatial Information Authority of Japan (2015)*.

All analytical codes (available in Supplemental Information 1) were written in R version 3.4.1 (*R Core Team, 2017*).

# RESULTS

## Effects of environmental variables

Predicted probability of bleaching increased with increasing values of thermal indices, including SST, DHM, DHW, and UV-B (Fig. 2). Predicted probability of bleaching decreased with DCW, water turbidity, and water depth (Fig. 2). Relationships between bleaching and historical SST variability and current speed were not significant, with 95% confidence intervals (CIs) ranging from negative to positive. Relationships between bleaching and monthly and weekly SST were positive, although the widths of the 95% CIs

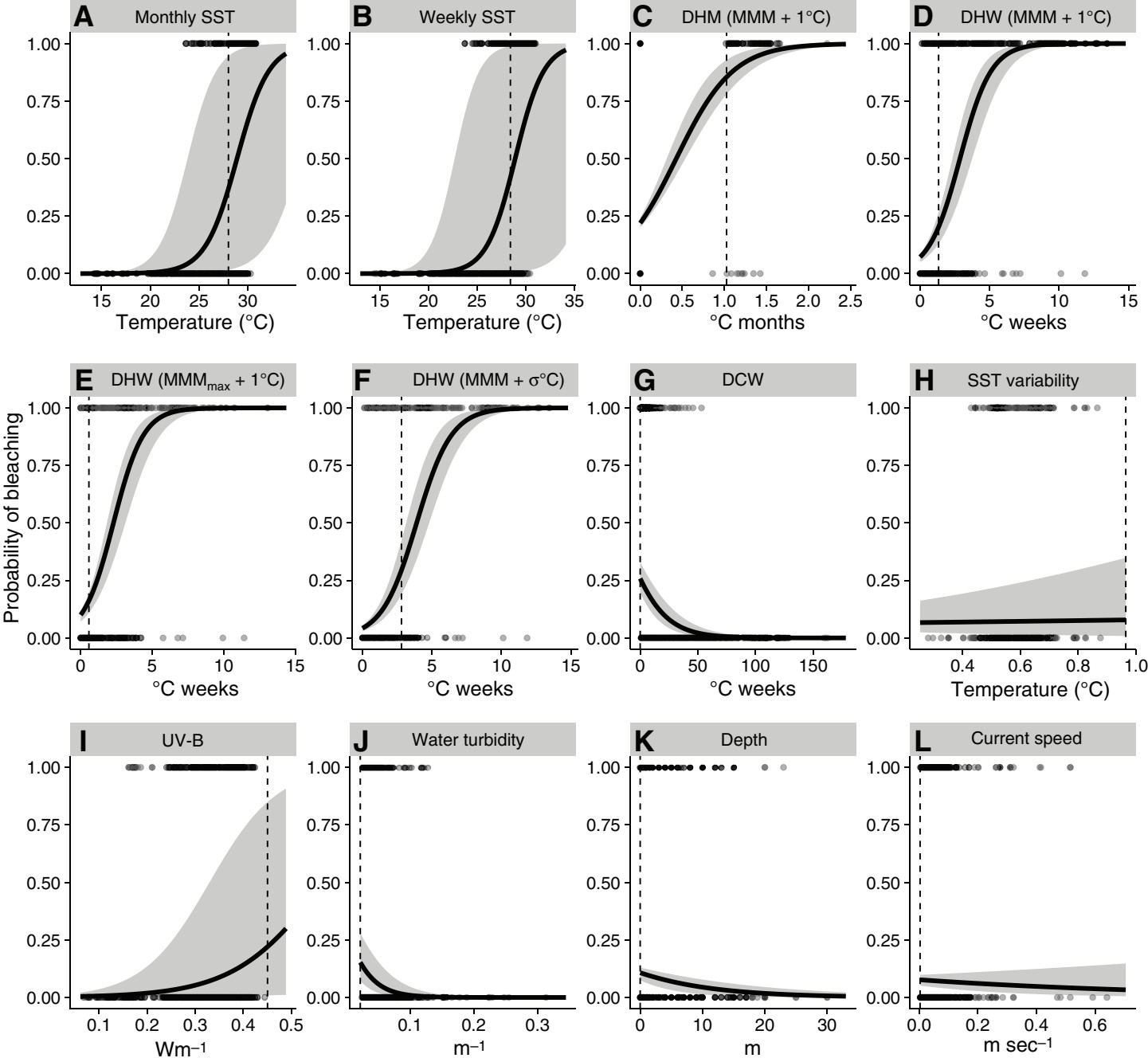

**Figure 2 Relationships between environmental variables and observed and predicted coral bleaching, obtained with univariate generalized linear models.** (A) Monthly sea-surface temperature (SST); (B) weekly SST; (C) degree heating month (DHM); (D) NOAA CRW degree heating week (DHW); (E) DHW using mean of the warmest monthly mean SST of each ear ($MMM_{max}$); (F) DHW using historical SST variability ($\sigma$) as filtering threshold; (G) degree cooling week (DCW); (H) historical SST variability; (I) UV-B; (J) water turbidity; (K) water depth; (L) current speed. Values of 1 and 0 represent bleaching and nonbleaching, respectively. Solid lines and gray areas indicate mean model fit and 95% confidence intervals, respectively. Dotted lines represent thresholds discriminating bleaching and nonbleaching, which were optimized by true positive rate and true negative rate (TPR–TNR) sum maximization (Table 2). See Table 2 for terminology.

suggest these variables are not reliable indices of coral bleaching. Alert thresholds for predicted bleaching were found to be lower than standard thresholds, except for DHM (Table 3).

**Table 3 Univariate prediction models of coral bleaching using thermal indices with optimized evaluation thresholds.**

| Model | Evaluation threshold (Bleaching alert threshold °C) | Predicted formula (for GLMs) |
|---|---|---|
| Monthly SST (GLM) | 0.377 ± 0.010 (28.01 °C) | logistic(−17.7 + 0.612·SST) |
| Monthly SST (RF) | 0.346 ± 0.010 | |
| Weekly SST (GLM) | 0.409 ± 0.012 (28.04 °C) | logistic(−19.7 + 0.680·SST) |
| Weekly SST (RF) | 0.309 ± 0.020 | |
| DHM (MMM + 1 °C) (GLM) | 0.855 ± 0.004 (1.02 °C) | logistic(−1.27 + 2.96·DHM) |
| DHM (MMM + 1 °C) (RF) | 0.454 ± 0.015 | |
| DHW (MMM + 1 °C) (GLM) | 0.208 ± 0.012 (1.33 °C) | logistic(−2.56 + 0.891·DHW) |
| DHW (MMM + 1 °C) (RF) | 0.129 ± 0.019 | |
| DHW ($MMM_{max}$ + 1 °C) (GLM) | 0.162 ± 0.009 (0.58 °C) | logistic(−2.2 + 0.958·DHW) |
| DHW ($MMM_{max}$ + 1 °C) (RF) | 0.268 ± 0.017 | |
| DHW (MMM + $\sigma_m$ °C) (GLM) | 0.292 ± 0.022 (2.81 °C) | logistic(−3.12 + 0.800·DHW) |
| DHW (MMM + $\sigma_m$ °C) (RF) | 0.196 ± 0.017 | |

Notes:
The optimized evaluation thresholds (mean ± SE) of the predicted probability of coral bleaching are shown with corresponding bleaching alert thresholds of thermal indices. The optimized formula for the predicted probability of bleaching is shown for GLM. logistic($x$) = 1/(1 + exp(−$x$)).
SST, sea-surface temperature; DHM, degree heating month; DHW, degree heating week; MMM, maximum of the monthly mean SST climatology; $MMM_{max}$, mean of the warmest monthly mean SST of each year; GLM, generalized linear model; RF, random forest.

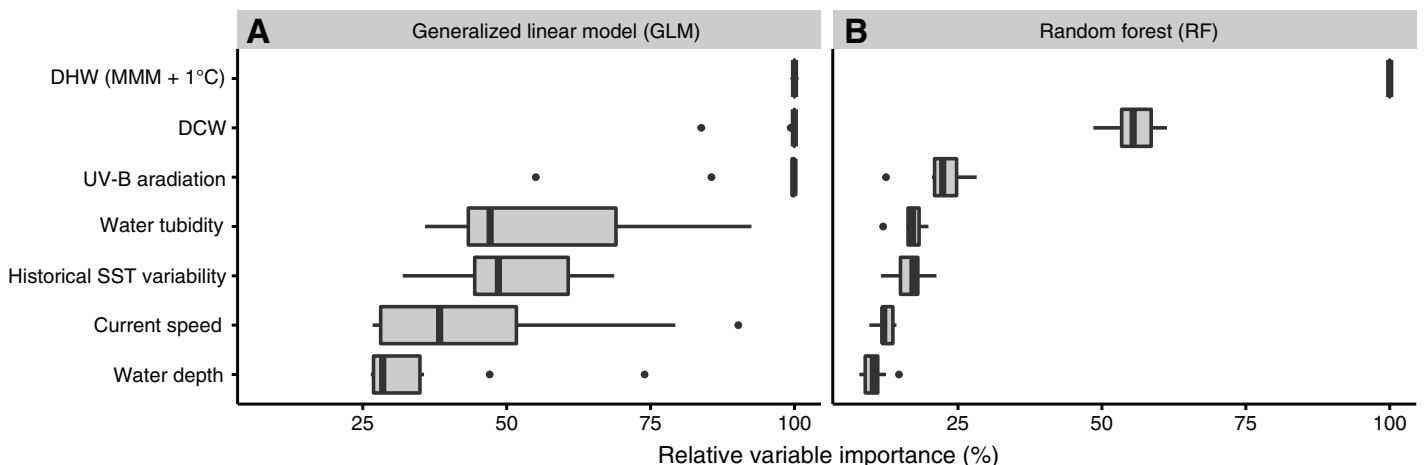

**Figure 3 Relative importance of environmental variables.** Under (A) generalized linear model (GLM) and (B) random forest (RF). DCW, degree cooling week; DHW, degree heating week; MMM, maximum monthly mean; SST, sea-surface temperature; UV-B, ultraviolet B.

Ranking of important variables was similar between GLM and RF (Fig. 3): the best explanatory variable was DHW (100% in both of GLM and RF), followed by DCW. UV-B, water turbidity, and historical SST variability also explained substantial variation in coral bleaching. The explanatory powers of historical SST variability and current speed were high, despite inconsistent relationships with coral bleaching (Fig. 2). Absolute variable importance differed between GLM and RF. In GLM, most variables explained more than

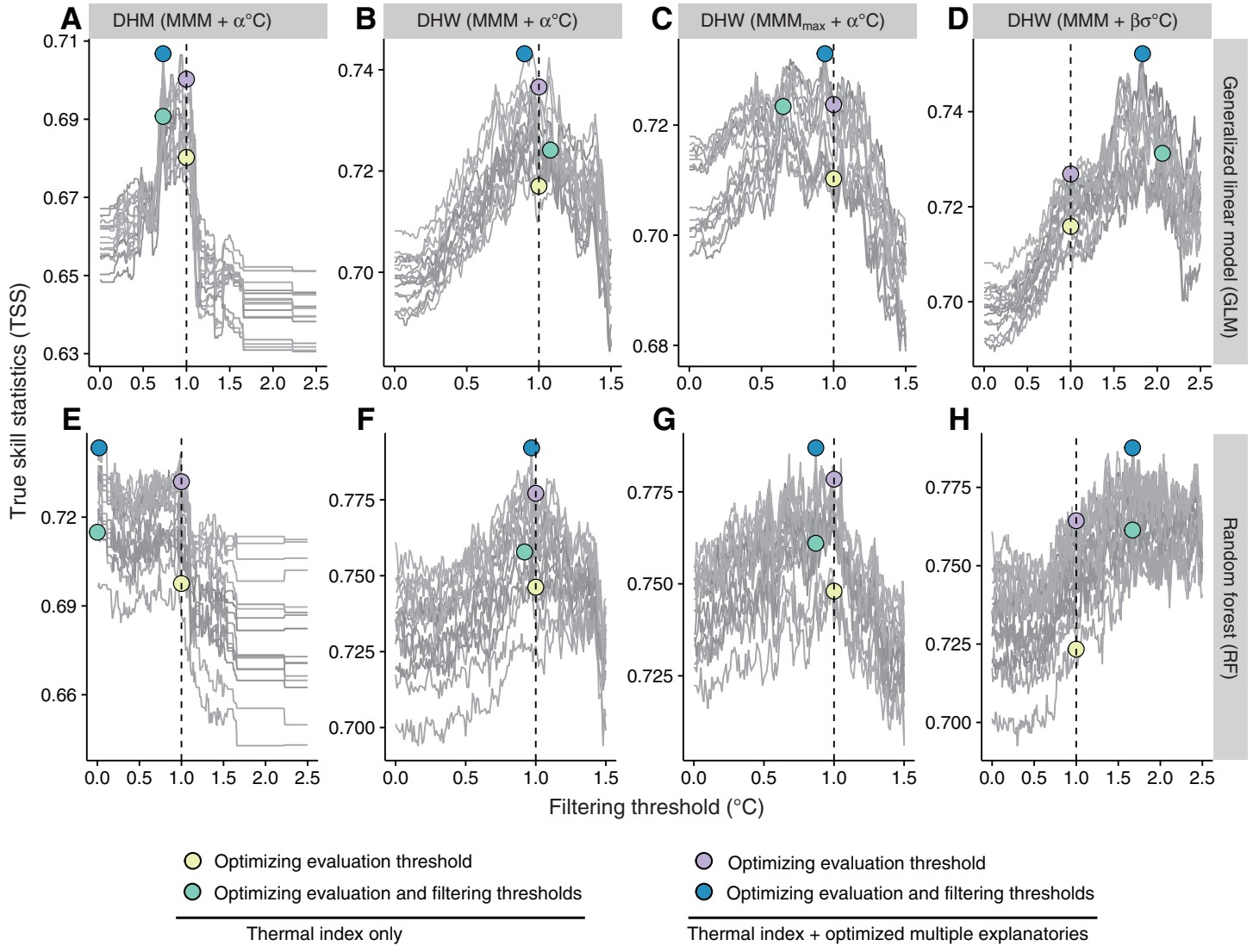

**Figure 4 Optimization of filtering thresholds.** Model predictive performance (true skill statistic, TSS) with varying filtering thresholds for four thermal indices under (A–D) a generalized linear model (GLM) and (E–H) a random forest (RF) (Tables 1 and 2). (A, E) DHM (MMM + α °C); (B, F) DHW (MMM + α °C); (C, G) DHW (MMM$_{max}$ + α °C); (D, H) DHW (MMM + β σ$_m$ °C). Individual gray lines represent each of the 15 combinations of environmental variables. DHM, degree heating month; DHW, degree heating week; MMM, maximum monthly mean. See Table 2 for terminology.

25% of variation in bleaching each, while in RF, only DHW, DCW, and UV-B explained more than 25% of variation in bleaching each.

## Optimization and assessment of filtering thresholds

Optimization of filtering thresholds improved the predictive performance of DHM and DHWs, although the improvement was small for GLM (Fig. 4). Improvement by the optimization was around 0.01 in TSS in GLM, while the improvement of DHW using the historical SST variability (σ$_m$) as the filtering threshold was 0.02–0.03 in TSS in RF.

We compared the predictive performance of all models including standard and optimized thermal indices, and the optimized set of explanatory variables (Fig. 5).

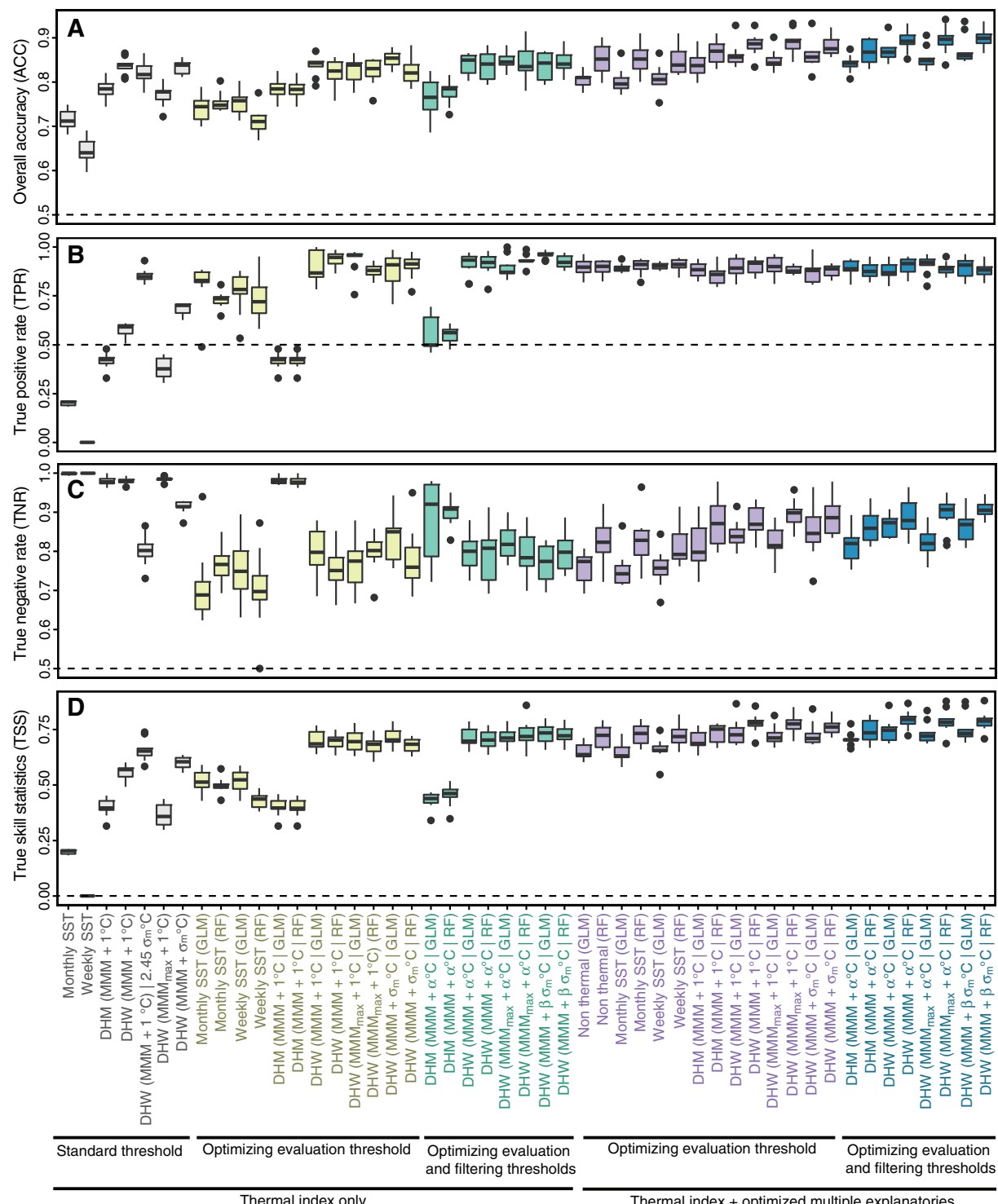

**Figure 5 Evaluation of models of coral bleaching.** (A) Overall accuracy (ACC); (B) True positive rate (TPR); (C) True negative rate (TNR); (D) True skill statistic (TSS). The label of a model indicates the index and filtering threshold, if present | abbreviation of statistical model. For example, DHW (MMM + β σ_m °C | RF) represents the random forest model, including DHW using MMM + β σ_m °C as the filtering threshold. See Table 2 for terminology and Tables 3–6 for optimized evaluation thresholds and filtering thresholds, and combinations of explanatory variables. DHM, degree heating month; DHW, degree heating week; GLM, generalized linear model; MMM, maximum monthly mean; RF, random forest; SST, sea-surface temperature.

**Table 4 Univariate prediction models of coral bleaching using thermal indices with optimized evaluation and filtering thresholds.**

| Model | Evaluation threshold (Bleaching alert threshold °C) | Filtering threshold | Predicted formula (for GLMs) |
|---|---|---|---|
| DHM (MMM + α °C) (GLM) | 0.464 ± 0.024 (0.611 °C) | α = 0.23 | logistic(−1.65 + 2.48·DHM) |
| DHM (MMM + α °C) (RF) | 0.369 ± 0.010 | α = 0.23 | |
| DHW (MMM + α °C) (GLM) | 0.213 ± 0.010 (2.07 °C) | α = 0.68 | logistic(−3.00 + 0.803·DHW) |
| DHW (MMM + α °C) (RF) | 0.207 ± 0.019 | α = 0.52 | |
| DHW (MMM$_{max}$ + α °C) (GLM) | 0.228 ± 0.006 (0.296 °C) | α = 0.64 | logistic(−2.64 + 0.892·DHW) |
| DHW (MMM$_{max}$ + α °C) (RF) | 0.174 ± 0.014 | α = 0.89 | |
| DHW (MMM + β·σ$_m$ °C) (GLM) | 0.134 ± 0.002 (1.58 °C) | β = 2.13 | logistic(−2.22 + 1.01·DHW) |
| DHW (MMM + β·σ$_m$ °C) (RF) | 0.146 ± 0.014 | β = 1.68 | |

Notes:
The optimized evaluation thresholds (mean ± SE) of the predicted probability of coral bleaching are shown with corresponding bleaching alert thresholds of thermal indices. The optimized formula for predicted probability of bleaching is shown for GLM. logistic($x$) = $1/(1 + \exp(−x))$.
SST, sea-surface temperature; DHM, degree heating month; DHW, degree heating week; MMM, maximum of the monthly mean SST climatology; MMM$_{max}$, mean of the warmest monthly mean SST of each year; GLM, generalized linear model; RF, random forest.

In models based only on a thermal index with the standard threshold, TNR was larger than TPR, indicating that high overall accuracy can result from effective identification of nonbleaching occurrences, despite ineffective identification of bleaching occurrences. TSS was indicative of both TPR and TNR. Among thermal indices with the standard threshold, DHW using historical SST variability ($σ_m$) as the bleaching alert threshold and DHW using $σ_m$ as the filtering threshold scored the best performances in TSS (0.60 and 0.65, respectively) (Fig. 5). Weekly SST had no prediction skill (TSS = 0) and the skill of monthly SST was low (0.20). TPR of DHW using historical SST variability as the bleaching alert threshold was the highest (0.85) among models, although the false-positive rate (1−TNR = 0.20) was also the highest. Using historical SST variability as the filtering threshold for DHW decreased the false-positive rate (0.09), but also decreased TPR (0.69). DHW using MMM$_{max}$ as the baseline climatology showed the lowest predictive performance (TSS = 0.37) among all DHW indices, with the lowest TPR (0.38).

Predictive performance was improved by optimizing the evaluation threshold (Fig. 5; Table 4). Under optimized evaluation thresholds, DHW with historical SST variability as the filtering threshold performed best (TSS = 0.72 for GLM) among the three types of DHW, although differences were small. Predictive performance of DHM was the lowest (TSS = 0.40). When both evaluation and filtering thresholds were optimized (Table 5), improvements in evaluation thresholds were only small. DHW with historical SST variability as the filtering threshold remained the highest performing index under optimization (TSS = 0.73 in both GLM and RF). Optimized filtering thresholds were less than 1 °C in all the models using one thermal index, using the optimized DHM or DHW other than DHW using historical SST variability ($σ_m$) as the filtering threshold (see also Fig. S3D for the distribution of $σ_m$).

The predictive skill of models with multiple explanatory variables was negligibly higher than that of models including only one thermal index. TSS increased by 0.1 at most in all

**Table 5 Multivariate prediction models of coral bleaching including thermal indices with optimized evaluation thresholds.**

| Model | Evaluation threshold | Optimized explanatory variables/predicted formula for GLMs |
|---|---|---|
| Non thermal (GLM) | $0.389 \pm 0.011$ | logistic($-1.37 - 0.112 \cdot c - 0.0341 \cdot d + 3.77 \cdot s + 7.95 \cdot u$) |
| Non thermal (RF) | $0.333 \pm 0.008$ | c, k, u, v |
| Monthly SST (GLM) | $0.387 \pm 0.011$ | logistic($-12.0 + 0.458 \cdot SST - 0.094 \cdot c + 4.10 \cdot s - 0.722 \cdot u$) |
| Monthly SST (RF) | $0.311 \pm 0.008$ | SST, c, k, s, u, v |
| Weekly SST (GLM) | $0.348 \pm 0.010$ | logistic($-17.2 + 0.667 \cdot SST - 0.084 \cdot c + 4.69 \cdot s - 3.56 \cdot u$) |
| Weekly SST (RF) | $0.322 \pm 0.005$ | SST, c, k, u, v |
| DHM (MMM + 1 °C) (GLM) | $0.387 \pm 0.010$ | logistic($1.31 + 3.02 \cdot DHM - 0.126 \cdot c - 0.026 \cdot d + 3.55 \cdot s + 2.99 \cdot u - 2.69 \cdot v$) |
| DHM (MMM + 1 °C) (RF) | $0.380 \pm 0.009$ | DHM, c, k, s, u, v |
| DHW (MMM + 1 °C) (GLM) | $0.326 \pm 0.005$ | logistic($-1.85 + 0.723 \cdot DHW - 0.053 \cdot c - 17.4 \cdot k + 2.29 \cdot s + 10.1 \cdot u - 3.57 \cdot v$) |
| DHW (MMM + 1 °C) (RF) | $0.365 \pm 0.008$ | DHW, c, k, s, u, v |
| DHW (MMM$_{max}$ + 1 °C) (GLM) | $0.325 \pm 0.006$ | logistic($-3.75 + 0.805 \cdot DHW - 0.065 \cdot c - 17.7 \cdot k + 1.18 \cdot s + 11.2 \cdot u$) |
| DHW (MMM$_{max}$ + 1 °C) (RF) | $0.395 \pm 0.007$ | DHW, c, k, u, v |
| DHW (MMM + $\sigma_m$ °C) (GLM) | $0.323 \pm 0.010$ | logistic($-1.99 + 0.688 \cdot DHW - 0.031 \cdot c - 18.3 \cdot k + 7.41 \cdot u - 3.11 \cdot v$) |
| DHW (MMM + $\sigma_m$ °C) (RF) | $0.393 \pm 0.008$ | DHW, c, k, s, u, v |

**Notes:**
c: DCW; d: depth; k: water turbidity; u: UV-B radiation; s: current speed; v: historical SST variability (see Table 2). The optimized evaluation thresholds (mean ± SE) of the predicted probability of coral bleaching are shown with corresponding bleaching alert thresholds of thermal indices. The optimized formula for predicted probability of bleaching is shown for GLM. logistic($x$) = $1/(1 + \exp(-x))$.
SST, sea-surface temperature; DHM, degree heating month; DHW, degree heating week; MMM, maximum of the monthly mean SST climatology; MMM$_{max}$, mean of the warmest monthly mean SST of each year; GLM, generalized linear model; RF, random forest.

models, except for the DHM model where TSS increased by 0.3 in both GLM and RF. Increases in predictive performance were mainly due to increases in TNR (i.e., reductions in false-positive rates; Fig. 5). Models including no thermal indices showed high prediction skill (TSS = 0.65 in GLM; TSS = 0.72 in RF). In models with multiple explanatory variables, differences in predictive performance between models with optimized evaluation thresholds and models with optimized evaluation and filtering thresholds were smaller than differences between GLM and RF models for most thermal indices. RF always performed better than GLM, with differences in TSS of 0.04 to 0.05. Although the TPR of GLM exceeded that of RF in most cases, the TNR of GLM was lower than those of RF, i.e., the risk of false positives was higher in GLM.

The RF model based on DHW with MMM + 0.97 °C filtering threshold, DCW, UV-B, water turbidity, historical SST variability, and current speed (Table 6) showed the best predictive performance (TSS = 0.79; TPR = 0.90; TNR = 0.89). Among the GLM, the model consisting of DHW with MMM + 1.83·$\sigma_m$ °C filtering threshold, DCW, UV-B, and turbidity showed the best predictive skill.

**Table 6 Multivariate prediction models of coral bleaching including thermal indices with optimized evaluation thresholds and filtering thresholds.**

| Model | Evaluation threshold | Filtering threshold | Optimized explanatory variables/predicted formula for GLMs |
|---|---|---|---|
| DHM (MMM + α °C) (GLM) | 0.388 ± 0.006 | α = 0.73 | logistic(1.53 + 2.47·DHM − 0.125·c + 3.47·s + 2.37·u − 3.17·v) |
| DHM (MMM + α °C) (RF) | 0.380 ± 0.009 | α = 0.02 | DHM, c, k, u, v |
| DHW (MMM + α °C) (GLM) | 0.354 ± 0.007 | α = 0.90 | logistic(−1.99 + 0.717·DHW − 0.048·c − 17.3·k + 2.39·s + 9.99·u − 3.74·v) |
| DHW (MMM + α °C) (RF) | 0.378 ± 0.010 | α = 0.97 | DHW, c, k, s, u, v |
| DHW ($MMM_{max}$ + α °C) (GLM) | 0.320 ± 0.008 | α = 0.94 | logistic(−3.73 + 0.789·DHW − 0.062·c − 19.0·k + 8.77·u) |
| DHW ($MMM_{max}$ + α °C) (RF) | 0.400 ± 0.008 | α = 0.87 | DHW, c, u, v |
| DHW (MMM + β·$\sigma_m$ °C) (GLM) | 0.336 ± 0.005 | β = 1.83 | logistic(−3.15 + 0.773·DHW − 0.053·c − 19.2·k + 8.77·u) |
| DHW (MMM + β·$\sigma_m$ °C) (RF) | 0.394 ± 0.006 | β = 1.67 | DHW, c, d, k, s, u, v |

**Notes:**

c: DCW; d: depth; k: water turbidity; u: UV-B radiation; s: current speed; v: historical SST variability (see Table 2). The optimized evaluation thresholds (mean ± SE) of the predicted probability of coral bleaching are shown with corresponding bleaching alert thresholds of thermal indices. The optimized formula for predicted probability of bleaching is shown for GLM. logistic($x$) = 1/(1 + exp(−$x$)).

SST, sea-surface temperature; DHM, degree heating month; DHW, degree heating week; MMM, maximum of the monthly mean SST climatology; $MMM_{max}$, mean of the warmest monthly mean SST of each year; GLM, generalized linear model; RF, random forest.

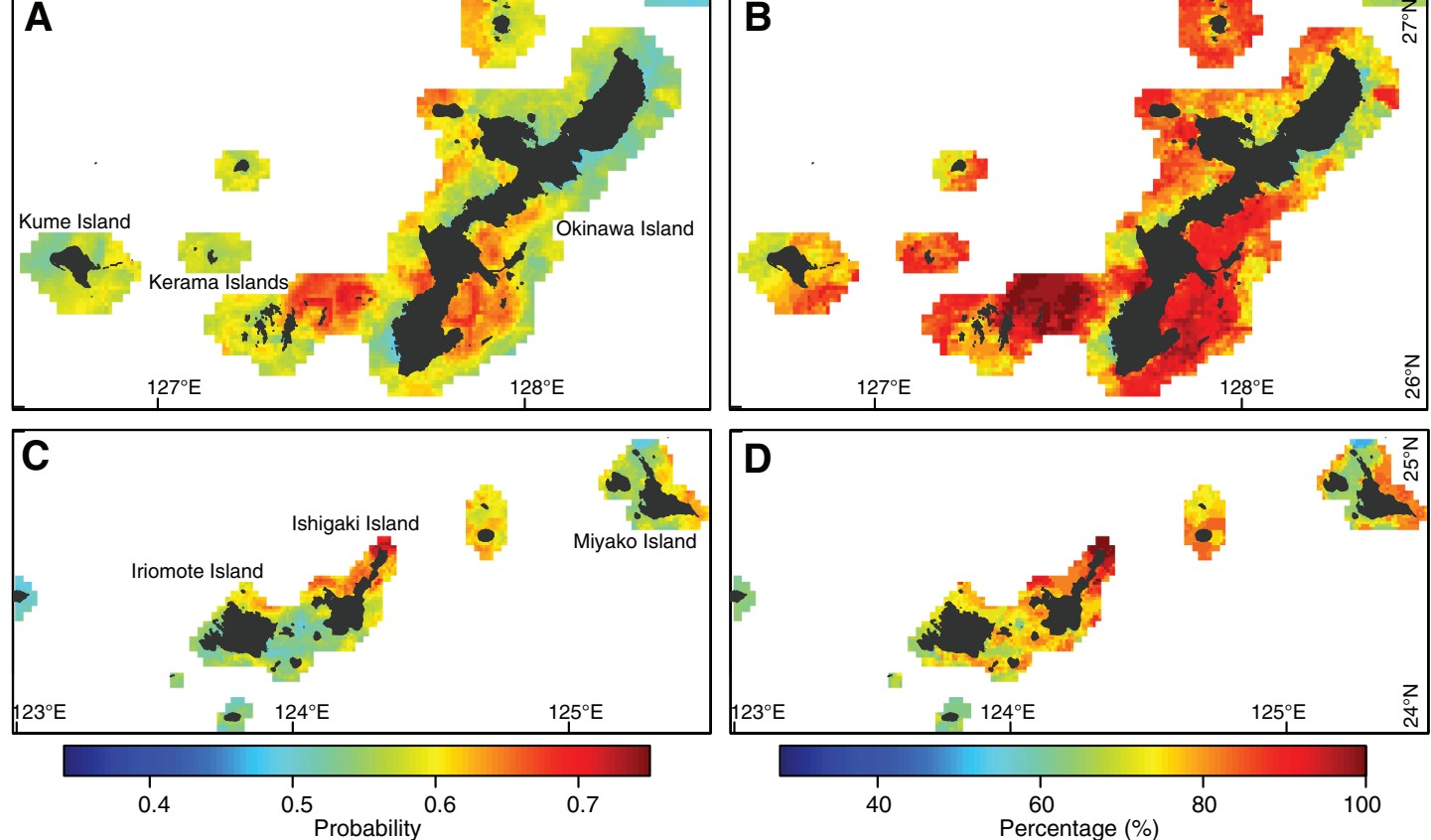

**Figure 6 Coral bleaching prediction under observed environmental conditions.** (A, B) Eastern Ryukyu Islands. (C, D) Western Ryukyu Islands. (A, C) Mean of bleaching probabilities in the warmest months in 2008–2010, 2013, and 2016. (B, D) Percentage of predicted bleaching frequencies for 2008–2010, 2013, and 2016. A value of 100% indicates bleaching in all years. The average results from 10 models built with cross-validations are shown. Japanese map is publicly available from the *Geospatial Information Authority of Japan (2015)* (http://www.gsi.go.jp/ENGLISH/index.html).

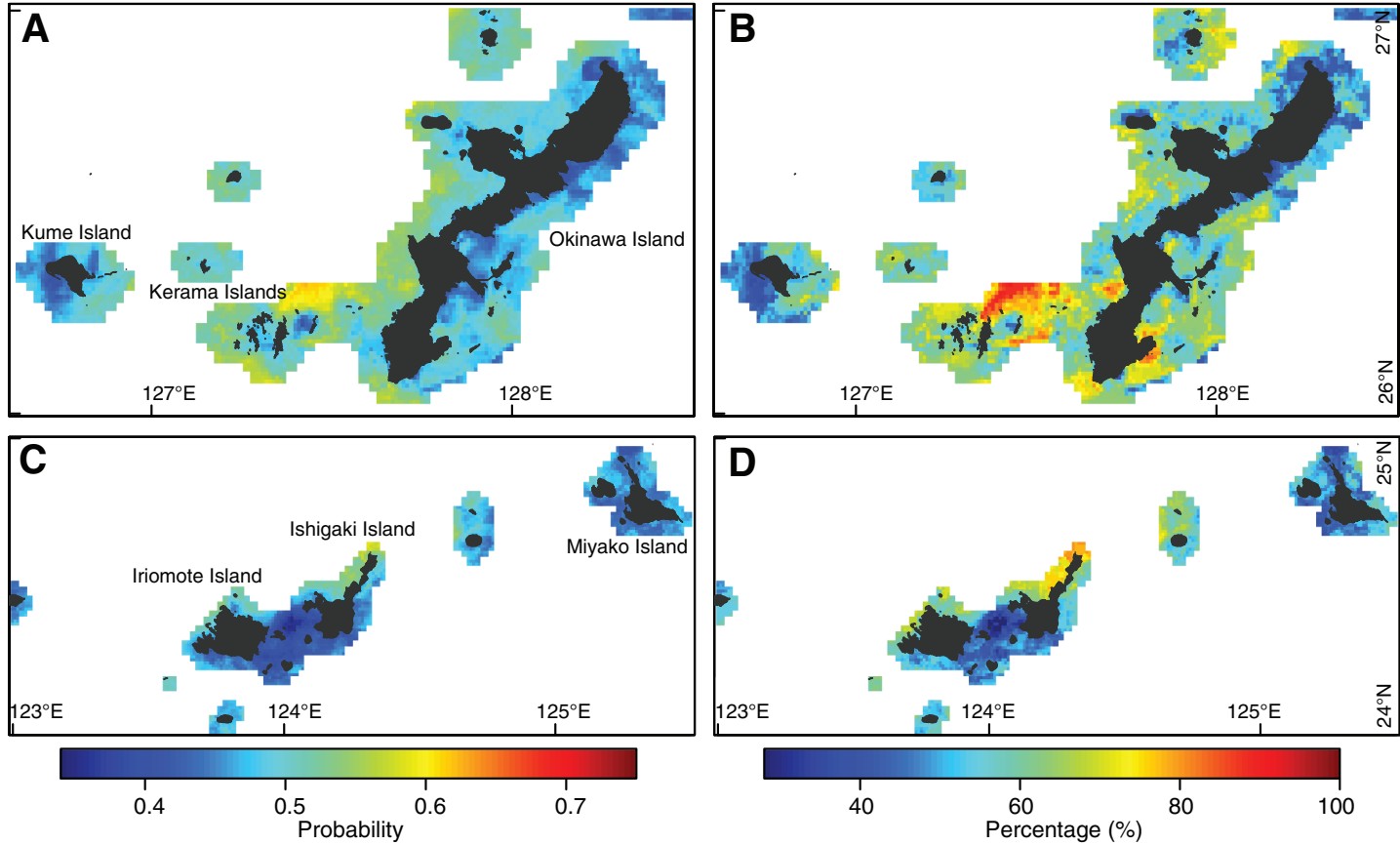

**Figure 7 Coral bleaching prediction under reduced ultraviolet B (UV-B) radiation due to screening.** Prediction under a 40% reduction in UV-B radiation due to a 40% increase in screening effect. (A, B) Eastern Ryukyu Islands. (C, D) Western Ryukyu Islands. (A, C) Mean of bleaching probabilities in the warmest months in 2008–2010, 2013, and 2016. (B, D) Percentage of predicted bleaching frequencies for 2008–2010, 2013, and 2016. A value of 100% indicates bleaching in all years. The average results from 10 models built with cross-validations are shown. Japanese map is publicly available from the *Geospatial Information Authority of Japan (2015)* (http://www.gsi.go.jp/ENGLISH/index.html).

## Predictions of coral bleaching

We predicted probabilities of coral bleaching in the main coral-habitable areas of Japan with the optimized best multivariate model of RF using DHW with MMM + 0.97 °C filtering threshold (Fig. 6). The mean predicted probability of bleaching ranged from 0.46 to 0.74 among areas. Spatial variation in the probability of bleaching was found in both the eastern (Fig. 6A) and the western (Fig. 6C) Ryukyu Islands. Hotspots with higher bleaching probabilities were found in the southeastern part of Okinawa Island, the eastern part of the Kerama Islands (Fig. 6A), and the northern part of Ishigaki Island (Fig. 6C). This resulted in bleaching in 2.5–5 (44–100%) of five years (2008–2010, 2013, and 2016; Figs. 6B and 6D).

Under reduced UV-B radiation, coral bleaching decreased in most areas, with particularly large decreases in bleaching hotspots (Figs. 7A and 7C). Predicted probabilities of bleaching ranged from 0.34 to 0.63 (Figs. 7A and 7C), and bleaching frequencies ranged from 28% to 92% (Figs. 7B and 7D). Decreases in probabilities of

bleaching of up to 0.24 were observed, resulting in a significant decrease in bleaching frequency of up to 56% (Figs. 7B and 7D). Bleaching in fewer than three out of five years occurred in most areas.

## DISCUSSION

### Optimizing coral bleaching models

The ability to predict coral bleaching was improved by optimizing thermal indices, particularly SST and DHWs. As in a previous study (*Donner, 2011*), we generally found lower TPR than TNR, but both TPR and TNR were improved by optimization. The sole contribution of optimizing the filtering threshold was small. However, optimizing the filtering threshold and the evaluation threshold while combining multiple environmental variables achieved large improvements in TPR and TNR (reaching ~0.9). We also found that cooling (DCW), UV-B, and screening (water turbidity) were important predictors of bleaching, particularly in RF models.

Our results are mostly consistent with those of *Donner (2011)*, who showed that DHW using historical SST variability as bleaching alert threshold had a higher TPR but a higher false-positive rate than NOAA CRW DHW, and that DHW using $MMM_{max}$ did not predict bleaching accurately but was suitable in equatorial zones. Because our study was conducted at a higher latitude, using $MMM_{max}$ resulted in lowest performance among the DHW indices, as expected. We also used historical SST variability as a filtering threshold, and this method showed the highest prediction skill among the models with only one thermal index. Historical SST variation ($\sigma_m$) may be a particularly effective predictor of bleaching in Japan, as variation was larger in our study area (ca. 0.56) than in the study area (ca. 0.25) of *Donner (2011)*. Indeed, southern islands in Japan are encircled by the strong Kuroshio boundary current flowing poleward, and tropical waters brought by the current can cause faster warming than the global average (*Wu et al., 2012*), leading to high levels of historical SST variation in the area.

Optimized filtering thresholds were smaller than 1 °C in GLM and RF models using only one thermal index. Previous studies have suggested that thermal stress not exceeding 1 °C can induce coral bleaching (*Brown, 1997*; *McWilliams et al., 2005*; *Kleypas, Danabasoglu & Lough, 2008*). Nevertheless, bleaching thresholds had not been statistically optimized before our study.

Random forest was an excellent method for predicting coral bleaching and could be used more widely in studies of coral ecology. The use of RF has much increased in ecological studies in the 10 years since the introduction of *Cutler et al. (2007)*. RF shows considerable potential for ecological analyses including classification, regression, and survival, due to its high predictive accuracy and its ability to model complex interactions among explanatory variables (*Cutler et al., 2007*).

Models with multiple environmental variables are becoming more popular, and show high explanatory power when modeling bleaching (*Maina et al., 2008*; *Yee, Santavy & Barron, 2008*; *McClanahan, Maina & Ateweberhan, 2015*; *Welle et al., 2017*). We found UV radiation to be an important explanatory factor for coral bleaching, consistent with previous studies (*Hoegh-Guldberg, 1999*; *Maina et al., 2008*, *2011*;

*McClanahan, Maina & Ateweberhan, 2015*). Other variables related to cooling (DCW; *Jones et al., 2017*) and screening (water turbidity; *West & Salm, 2003*; *Oliver, Berkelmans & Eakin, 2009*; *Maina et al., 2011*; *Oxenford & Vallés, 2016*) also explained variation in the occurrence of coral bleaching. Small-scale topographic variables, including water depth, are known to reduce thermal stress on corals (*West & Salm, 2003*; *Oliver, Berkelmans & Eakin, 2009*). Strong winds may also reduce bleaching risk (*Maina et al., 2008*, *2011*; *Yee, Santavy & Barron, 2008*; *McClanahan, Maina & Ateweberhan, 2015*; *Welle et al., 2017*), but the importance of this variable could not be evaluated in our study due to high correlations with thermal indices. Our predictions may be negatively affected by environmental variation at small temporal and spatial scales that has not been adequately included in our study. For example, small-scale water flow may improve the resistance of corals to bleaching (*Nakamura & van Woesik, 2001*), but the 8 km resolution of current speed in our study is too coarse to represent such effects. The microstructure of the sea floor at the meter scale may also be related to local water flow or shading of corals (*West & Salm, 2003*; *Oliver, Berkelmans & Eakin, 2009*), but was not incorporated in our study.

Bleaching responses and thermal thresholds vary among coral species (*Maynard et al., 2008*; *Yee, Santavy & Barron, 2008*; *Guest et al., 2012*; *Harii et al., 2014*; *McClanahan, 2014*). Branching corals of *Acropora* and *Pocillopora* spp. are more susceptible to thermal stress than massive corals such as *Porites* spp. (*Maynard et al., 2008*; *Yee, Santavy & Barron, 2008*; *Guest et al., 2012*; *Harii et al., 2014*; *McClanahan, 2014*). Thermal tolerance increases with repeated excessive thermal stress (*Brown et al., 2002*; *Maynard et al., 2008*; *Guest et al., 2012*), highlighting the potential of corals to adapt to thermal stress (*Brown et al., 2002*; *Maynard et al., 2008*; *Guest et al., 2012*). However, the effects of past thermal conditions have not been fully explained with historical SST variability in our analysis or in previous studies (*Donner, 2011*; *McClanahan, Maina & Ateweberhan, 2015*). Variation in thermal tolerance can result from interspecific differences (*Maynard et al., 2008*; *Guest et al., 2012*; *Harii et al., 2014*; *McClanahan, 2014*) in acclimation, genotypes, and epigenetics of host corals and symbiotic algae (*Palumbi et al., 2014*; *Torda et al., 2017*). The effects of such differences remain poorly known and should be prioritized for further research.

## Coral bleaching in Japan and reef management

Four studies in the Japanese region have analyzed coral bleaching occurrences with temperature anomalies and DHW at coarse resolutions (>50 km; *Strong et al., 2002*; *Harii et al., 2014*; *Kayanne, 2017*; *Kayanne, Suzuki & Liu, 2017*). Coarse DHW captures regional trends in the onset of coral bleaching, although it fails to predict bleaching within smaller reefs (*Strong et al., 2002*; *Harii et al., 2014*). Indeed, our bleaching predictions for the Ryukyu Islands with DHW at 1 km resolution exhibited a TPR of 0.58, compared to a value of 0.44 calculated from the tables of *Kayanne (2017)*. However, the improvement in predictive performance may result from the use of high-resolution temperature data, or the optimization of thermal thresholds of DHW. Further studies are required to establish the importance of high-resolution temperature data for predicting coral bleaching in Japan.

The high performance of our bleaching model at 1 km resolution has practical implications for the local and regional management of coral reefs. Our predictions revealed high frequencies of coral bleaching in many parts of the Ryukyu Islands. However, predictions of bleaching frequency were based on the lowest levels of bleaching severity; hence, further analyses may be required to establish full distributions of bleaching frequencies according to levels of severity.

Practical management to reduce the risk of coral bleaching should include control of coastal water turbidity (*Fabricius, 2005*). Increases in water turbidity by terrestrial runoff may decrease the resistance (*Wooldridge & Done, 2009*) or resilience (*Hongo & Yamano, 2013*) of corals to bleaching. Turbid coastal regions may provide refuges from climate warming due to limited increases in temperature and solar radiation (*Cacciapaglia & van Woesik, 2016*). However, coastal turbidity may increase the incidence of coral diseases and promote the growth of competing algae (*Fabricius, 2005*). Consequently, coastal turbidity should be carefully managed.

Reducing UV radiation may reduce bleaching risk and may constitute a powerful adaptive measure against climate warming. In the Onna Village of the Ryukyu Islands, in situ reduction of UV radiation with no increase in water turbidity has already been tested (*Okinawa Prefecture, 2017*). Reduction of UV radiation by 30–44% with large fishery nets resulted in a survival rate of 80% in cultured coral colonies in the summer of 2016, when the most severe thermal stress was recorded in the 2004–2016 study period (*Kayanne, Suzuki & Liu, 2017*). Reduction in UV radiation was similar to that used in our study (40%), so our predictions could provide a quantitative basis for future reef management in this area.

## CONCLUSION

Predictive performance of coral bleaching models can be improved by the use of optimized thresholds, multiple environmental influences, and multiple modeling methods. Both high-resolution modeling and observational records (i.e., the Sango Map Project) enabled high performance of bleaching predictions (*Oliver, Berkelmans & Eakin, 2009*). We provide a template for selecting appropriate indices to predict bleaching, and our research methods could be applied to coral-habitable areas globally. Our high-resolution predictions also provide a quantitative basis for the local and regional management of coral reefs (*West & Salm, 2003*). Although corals are suffering from high risks of bleaching globally, our study suggests that reducing UV radiation may be a key tool to improve coral resilience in the coming decades. Holistic bleaching models operating at finer spatial resolutions and incorporating variations in intrinsic thermal tolerance, historical effects of previous thermal impacts, and local environmental conditions should be the focus of future research. Such models will become indispensable as the effects of local and global stressors on corals continue to increase.

## ACKNOWLEDGEMENTS

We thank the 244 participants in the Sango Map Project who provided valuable data on coral bleaching events.

### Funding

This work was supported by the SOUSEI and TOUGOU Program of the Ministry of Education, Culture, Sports, Science, and Technology in Japan (MEXT) and by the Environment Research and Technology Development Fund (S15) of the Ministry of the Environment, Japan. The funders had no role in study design, data collection and analysis, decision to publish, or preparation of the manuscript.

### Grant Disclosures

The following grant information was disclosed by the authors:
SOUSEI and TOUGOU Program of the Ministry of Education, Culture, Sports, Science, and Technology in Japan (MEXT).
Environment Research and Technology Development Fund (S15) of the Ministry of the Environment, Japan.

### Competing Interests

The authors declare that they have no competing interests.

### Author Contributions

- Naoki H. Kumagai conceived and designed the experiments, analyzed the data, contributed reagents/materials/analysis tools, prepared figures and/or tables, authored or reviewed drafts of the paper.
- Hiroya Yamano conceived and designed the experiments, authored or reviewed drafts of the paper, Hiroya Yamano and the committee of the Sango Map Project constructed the system of the Sango Map Project.

### Data Availability

  The raw datasets have been provided as Supplemental Dataset Files.

### Supplemental Information

Supplemental information for this article can be found online at http://dx.doi.org/10.7717/peerj.4382#supplemental-information.

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
