# Peer review of "High-resolution modeling of thermal thresholds and environmental influences on coral bleaching for local and regional reef management"

_PeerJ, doi:10.7717/peerj.4382_

## Round 0.1 · original submission · Major Revisions

Dear Authors,

I have received two good reviews of your paper. Both suggest revision. It seems that the major issue is in presentation - the use of the English language. Please, make sure that you attend to the Referee's comments and get your language streamlined. Last but not least: please adhere strictly to the formats of PeerJ when resubmitting.

Sincerely
L Mucina, Editor

Reviewer 1 ·

Basic reporting

The manuscript “High-resolution modeling of thermal thresholds and 1 multiple environmental influences on coral bleaching
for regional and local reef managements” by Kamagai and Yamano provides a new and useful modeling approach for coral bleaching integrating citizen science, sophisticated statistical and machine algorithms and satellite data. Introduction and background shows well the context of the research. Literature is well referenced and relevant. Structure conforms to PeerJ standards, discipline norm, or improved for clarity. Figures are relevant, high quality, well labeled and described. Raw data and some codes are supplied. To improve the manuscript to be accepted authors must improve professional English language used throughout. I also recommend authors making available all R codes (including figures/maps plotting).
The manuscript is relevant and may be accepted to publication after the minor modifications suggested bellow.

Experimental design

The original primary research is within Scope of the journal.
Research question is well defined, relevant and meaningful. It is stated how the research fills an identified knowledge gap.
Rigorous investigation performed to a high technical and ethical standard.
Methods described with sufficient detail and information to replicate but I suggest including all codes (including for plotting all figures).
Authors chose GLM and Random Forest for modeling bleaching but not justify why these algorithms were chosen. Please add a sentence explaining the motivation for this choice.

Validity of the findings

Data is robust, statistically sound and controlled.
Conclusions are well stated, linked to original research question and limited to supporting results.

Additional comments

Abstract:
Authors refer coral reefs as corals. Corals are animals and coral reefs the environment.

Material and Methods:
The first paragraph of this section should be considerable shortened.
In line 110 I found: “An observer who attempted to submit an observational record to the Sango Map Project was requested to provide the following information as mandatory fields for quality control”. All this information is really necessary or authors can site the monitoring program?

Lines 140-146: this section should be considerable shortened

Lines 215-217: It is not clear why authors used this data/information. If it is relevant for models building please add a sentence explaining.

Line 233-249:
Why authors chosen RF and GLM? Please add a sentence explaining the motivation for this choice.
What R package was used to perform GLM, please add a citation.
There are some assumptions for GLM, what about RF? Please add a sentence explaining if there are some or not.

Results:
Line 299: “worst” does not sound scientific for variable evaluation. Please add a more technical term such “did not explain well…”. Please see line 367 in Discussion section, this comment extends also for the term “good”.

Table 3 – It is uncommon to see tables divided by letters. I suggest to reformat this table (all lines should have values/observations for each column) or to split in four tables.

Discussion:
Lines 385-387: I really think using RF in to model coral bleaching is very interesting and innovative, please add a sentence explaining how RF is being using for ecology modeling.

Reviewer 2 ·

Basic reporting

Kumagai and Yamano took a bold move to challenge NOAA’s predictive model of coral bleaching. By optimizing thermal thresholds and using different environmental parameters and modeling methods, the authors describes the potential of their high-resolution bleaching predictive model to improve local reef management. This interesting insight may be very helpful for small coastal countries where potential bleaching incidents may be overlooked by general models. One concern is that the screening experiment is considered a trial and should not be emphasized in the abstract. The paper is generally easy to follow. However, it will also certainly benefit from proof reading to correct the numerous typographical and grammatical errors. Tables and figures are important and it is critical to lead your audience on how to read or use the information.

Abstract:

1. Abstract can be more concise and seems to be very similar to the conclusion section of the paper.

Introduction:

1. The authors need to have supporting citations even for general information. For example, in P.3, line 41: “Extreme rising sea temperature … reef-building corals (REF, REF).

2. P.4, line 78: “Observations in local areas of research interest … Database. Can the authors name some areas covered?

3. P.4, line 80: Can the authors name some areas where records are limited?

4. P.5, line 100: Figures and tables should not be mentioned in Introduction.

Materials and methods

1. P.7, line 149: Table 1 is an important information. Authors should describe and introduce Table 1 appropriately in the text such as
“Steps to analysed the data were summarized in Table 1”.

2. P. 10, line 212. Similarly, for table 2, authors should explain how to use and read table

Conclusion:

1. Conclusion is very similar to abstract, authors should consider rephrasing either segments.

Table and figures:

1. Table 1: Caption should be more refined and proper. For those without references, are those steps created by the authors? If so, please include ‘This study (2017)”.

2. Table 2: For those terminologies without references, are those terms created by the authors? If so, please include ‘This study (2017)”.

Experimental design

Materials and methods

1. P.6, line 112 to 124: What about coral area surveyed?

2. P.10, line 204: Were the climatologic data from July to September obtained from 1997 to 2016?

3. P.13, line 270. Suggest to review the sentence to “In total, we evaluated 22,650 and 36,150 models … RF model respectively”.

Discussion:

1. P.20, line 443-449: “Our study showed that reducing UV radiation by increasing screening significantly reduced bleaching risk as … warming”. An actual study of coal screening was not described in materials and methods. Authors should include the description of the experiment if they want to include this point.

Validity of the findings

Results:

1. Results can be more qualitative than descriptive. The authors have all the numbers in tables and figures, use them to reinforce the results segment. For example, the improve scores can also be included in results segment.

Additional comments

Abstract:

1. P.1, line 3: Suggest to replace “managements’ with ‘management

2. P.2, line 17: Suggest to remove ‘living’

3. P.2, line 18-20: I don’t understand this sentence. Can the authors try to rephrase this or combine with the previous sentence?

4. P.2, line 31-32: Suggest to review the sentence to “Prediction based on the best explanatory model revealed that coral reefs in Japan are experiencing bleaching in many areas recently. A practical method to reduce bleaching frequency by screening UV radiation was also demonstrated in this paper.”

Introduction:

1. P.3 line 47: Suggest to revise phrase to “Degree Heating Weeks (DHW).

2. P.3, line 50: Citing one significant paper from the author should suffice.

3. P.4, line 70-75: Suggest to revise the sentence to “Furthermore, there are potentially interacting environmental stressors such as ultraviolet (UV) radiation (Hoegh-Guldberg, 1999; West & Salm 2003; Maina et al. 2008; Yee at al., 2008) and variables such as water turbidity (REF), topography of the sea floor (REF), … that can affect coral bleaching.

Materials and methods

1. P.9, line 195: Can the authors explain what is “MODIS-Aqua and Terra”?

2. P.13, line 270. Suggest to review the sentence to “In total, we evaluated 22,650 and 36,150 models … RF model respectively”.

Discussion:

1. P.20, line 429: Replace “coral bleaches’ with ‘coral bleaching’.

---

## Round 0.2 · accepted · Accept

Congratulations! I concur with both referees of your revised manuscript and suggest the acceptance of your paper.

Reviewer 1 ·

Basic reporting

Authors had greatly improved the manuscript after revision. I recommend this manuscript for publication in PeerJ in the present form.

Experimental design

no comment

Validity of the findings

no comment

Additional comments

no comment

Reviewer 2 ·

Basic reporting

No comment

Experimental design

No comment

Validity of the findings

No Comment